# Cost-effectiveness of a "treat-all" strategy using Direct-Acting Antivirals (DAAs) for Japanese patients with chronic hepatitis C genotype 1 at different fibrosis stages

**Riichiro Suenaga**[1], **Machi Suka**[2], **Tomohiro Hirao**[3], **Isao Hidaka**[4], **Isao Sakaida**[4], **Haku Ishida**[5]*

**1** Japanese Red Cross Yamaguchi Hospital, Yamaguchi, Yamaguchi, Japan, **2** Department of Public Health and Environmental Medicine, The Jikei University of Medicine, Minato-ku, Tokyo, Japan, **3** Department of Public Health, Faculty of Medicine, Kagawa University, Kita-gun, Kagawa, Japan, **4** Department of Gastroenterology & Hepatology, Yamaguchi University Graduate School of Medicine, Ube, Yamaguchi, Japan, **5** Department of Medical Informatics & Decision Sciences, Yamaguchi University Graduate School of Medicine, Ube, Yamaguchi, Japan

* hishida@yamaguchi-u.ac.jp

**Data Availability Statement:** All relevant data are within the manuscript and its Supporting Information files.

## Abstract

### Aim

To evaluate the cost-effectiveness of therapeutic strategies initiated at different stages of liver fibrosis using three direct-acting antivirals (DAAs), sofosbuvir-ledipasvir (SL), glecaprevir-pibrentasvir (GP), and elbasvir plus grazoprevir (E/G), for Japanese patients with chronic hepatitis C (CHC) genotype 1.

### Methods

We created an analytical decision model reflecting the progression of liver fibrosis stages to evaluate the cost-effectiveness of alternative therapeutic strategies applied at different fibrosis stages. We compared six treatment strategies: treating all patients regardless of fibrosis stage (TA), treating individual patients with one of four treatments starting at four respective stages of liver fibrosis progression (F1S: withholding treatment at stage F0 and starting treatment from stage F1 or higher, and three successive options, F2S, F3S, and F4S), and administering no antiviral treatment (NoRx). We adopted a lifetime horizon and Japanese health insurance payers' perspective.

### Results

The base case analysis showed that the incremental quality-adjusted life years (QALY) gain of TA by SL, GP, and E/G compared with the strategies of starting treatments for patients with the advanced fibrosis stage, F2S, varied from 0.32 to 0.33, and the incremental cost-effectiveness ratios (ICERs) were US$24,320, US$18,160 and US$17,410 per QALY, respectively. On the cost-effectiveness acceptability curve, TA was most likely to be cost-effective, with the three DAAs at the willingness to pay thresholds of US$50,000.

**Funding:** TH, HI and MS received grant from Ministry of Health, Labour and Welfare of Japan. Grant number is K8026002. https://mhlw-grants. niph.go.jp/ The funders had no role in study design, data collection and analysis, decision to publish, or preparation of the manuscript.

**Competing interests:** Isao Hidaka received honoraria from AbbVie and Gilead and received research funding from AbbVie. Isao Sakaida received honoraria from Gilead and received research funding from AbbVie. These don't alter our adherence to PLOS ONE policies on sharing data and materials.

**Abbreviations:** HCV, Hepatitis C virus; CHC, Chronic hepatitis C; CEA, cost-effectiveness analysis; DAA, direct antiviral agent; ICER, incremental cost-effectiveness ratio; QALY, quality-adjusted life years; SL, sofosbuvir-ledipasvir; GP, glecaprevir-pibrentasvir; E/G, elbasvir and grazoprevir.

## Conclusions

Our results suggested that administration of DAA treatment for all Japanese patients with genotype 1 CHC regardless of their liver fibrosis stage would be cost-effective under ordinary conditions.

## Background

The burden of hepatitis C virus (HCV) infections is a global problem, with about 80 million individuals currently estimated to have active viremic HCV infection worldwide [1]. The 69th World Health Assembly in 2016 adopted the first 'Global Health Sector Strategy on Viral Hepatitis' to eliminate viral hepatitis by 2030 as a public health threat [2].

In Japan, the rate of HCV prevalence is thought to have declined, since improved detection of HCV in blood transfusions has reduced the number of new infections, but 1.0 to 1.5 million individuals are still in an actively viremic state [3]. Moreover, the majority of patients were infected with HCV more than 20 years ago, making them vulnerable to the progression of fibrosis and advanced liver disease associated with persistent viremia. Therefore, comprehensive measures to combat hepatitis have been implemented including the public subsidy program for hepatitis treatment, which covers newly approved antiviral agents even in Japan.

The results of therapy for chronic hepatitis C (CHC) have been markedly improved by direct-acting antivirals (DAAs), and nearly all CHC patients treated with these agents can achieve a sustained virological response (SVR). Although DAAs are still expensive in Japan, several studies have concluded that they are cost-effective for patients with CHC due to their higher effectiveness [4–7].

Besides, the first edition of the Japanese guidelines for the treatment of patients with chronic hepatitis C recommended that interferon-based antiviral treatment should be applied on a priority basis for the patients with significant fibrosis (METAVIR score F2 or F3) or cirrhosis (METAVIR score F4). However, it did not actively recommend such treatment for the low-risk HCC group made up of non-elderly patients without advanced fibrosis, due to the limited efficacy and high side effect profile associated with interferon-based antiviral treatment as described in 2012, the year the first guideline was published [8].

Due to the availability of new DAAs that are more effective and better-tolerated, the recent clinical guidelines for HCV patients published by the Japan Society of Hepatology state that all HCV-infected patients except for decompensated cirrhosis patients should be considered eligible for antiviral therapy. Moreover, due to the high effectiveness and safety of current interferon-free DAAs, the treatment should be introduced at an early stage for the aforementioned, non-elderly HCC patients without advanced fibrosis, who are at low risk. Similarly, the American Association for the Study of Liver Diseases (AASLD) and the European Association for Study of the Liver (EASL) recommended that treatment with DAAs be considered for all treatment-naïve and treatment-experienced patients irrespective of their fibrosis stages, including patients with compensated or decompensated chronic liver disease due to HCV, provided they have no contraindications to treatment [9, 10]. As a result of these measures, viral elimination has been successfully progressing [11, 12].

On the other hand, facing the cost burden under the universal health coverage system, we have to reveal the cost-effectiveness of the DAA treatment for HCV-infected patients without contraindications [13].

Several studies that confirmed the cost-effectiveness of the DAA treatments for HCV-infected patients were published in other countries [14]. Besides, previous studies have also revealed the favorable cost-effectiveness of DAA treatment for the patients regardless of their genotype [15], adolescent patients [16], patients with HIV infection [17], and the universal screening program for HCV followed by DAAs treatments for the general population and sub-populations including prisoners and injecting drug users [18]. These results led to the recommendations by the AASLD and EASL and should guide the judgments of the health policymakers. Remarkably, the latest DAA regimens have a similar profile of effectiveness and safety, therefore, a difference in their costs mostly impacts the cost-effectiveness, and these results would give some evidence for the selection of the specific DAA treatment.

So far, however, we could not find any studies evaluated the cost-effectiveness of the DAA treatment for all patients with HCV infection, including those of less or no fibrosis under Japanese circumstances.

Therefore, in the present study we performed a cost-effectiveness analysis to compare the impacts of different treatment approaches for Japanese patients with chronic hepatitis C genotype 1: a strategy of treating all patients irrespective of their fibrosis stages; and a strategy of waiting and initiating treatment at the more advanced stages of liver fibrosis. Three currently available interferon-free DAA combinations—sofosbuvir-ledipasvir (SL), glecaprevir-pibrentasvigr (GP), and elbasvir (EBV) plus grazoprevir (GZR) (E/G)—were used as the treatment agents.

This study's object was to reveal the cost-effectiveness of these DAAs treatment for all treatment naïve patients with chronic hepatitis C genotype 1 irrespective of their fibrosis stage, to provide the policymakers with the cost perspective of national measures toward HCV hepatitis elimination.

## Methods

### Model for CEA evaluation

A state-transition Markov model for CEA was constructed based on natural history models of chronic hepatitis C. It consisted of five chronic hepatitis states classified by METAVIR fibrosis scores reflecting the progression of fibrosis stages: F0 (no fibrosis), F1 (portal fibrosis without septa), F2 (portal fibrosis with few septa), F3 (numerous septa without cirrhosis) and F4 (liver cirrhosis), as well as decompensated liver cirrhosis (decLC) and hepatocellular carcinoma (HCC). The Markov model was further combined with a post-treatment model that included both the antiviral treatment state and the post-antiviral treatment states, and also both the liver transplantation state and the post-liver transplantation state (Fig 1). Age- and gender-specific general population mortality rates from the 2017 life table published by the Ministry of Health, Labor and Welfare in Japan were considered for each state. The model was designed to output discounted costs and quality-adjusted life-years (QALYs) through the patients accrue the corresponding the cost and QALY of the health state for each cycle over a lifetime.

### Transition probabilities

We applied two series of progression rates of fibrosis obtained from a meta-analysis by Thein et al., which included two studies on Japanese subjects [19]. In the first system, rates are estimated using the meta-regression model developed in their analysis, and in the second system they are estimated by a random-effects model for patients whose infection periods was more than 20 years. The parameters in the meta-regression model were the duration of HCV infection (years), study design, proportion of males, proportion of genotype 1, age at HCV infection, proportion of excess alcohol cosumption, and risk of HCV acquisition. We calibrated

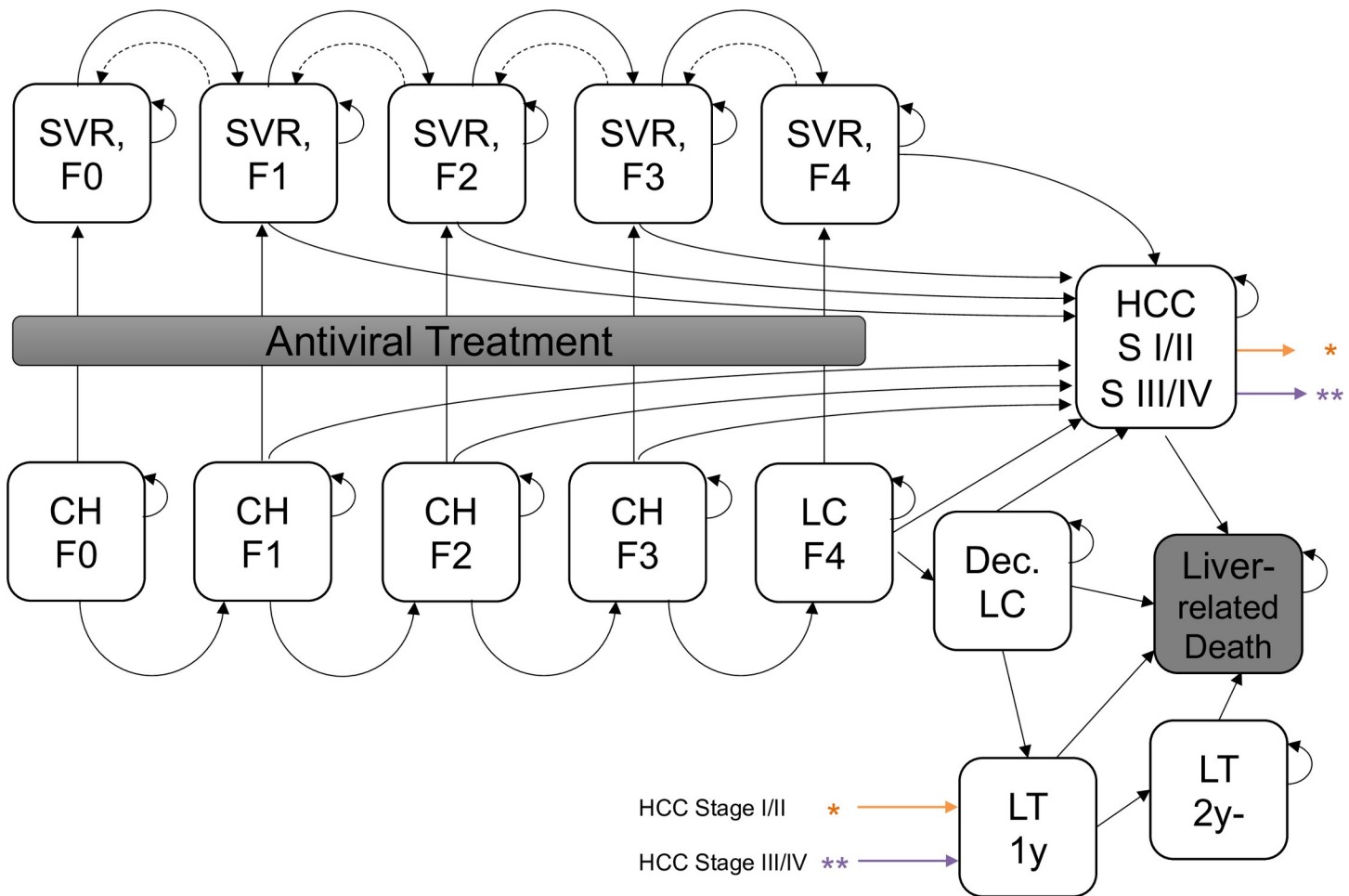

**Fig 1. Model scheme for chronic hepatitis C.** CH: Chronic hepatitis; LC: Compensated cirrhosis; Dec LC: Decompensated cirrhosis; HCC S I/II: Hepatocellular carcinoma stage I or II; HCC SIII/IV: Hepatocellular carcinoma stage III or IV; LT 1y: Liver transplantation; LT 2y-: Post-liver transplantation; SVR: Sustained virologic response. "Liver-related Death" represents the disease-specific mortality associated with having decompensated cirrhosis, liver transplant, or hepatocellular carcinoma. Patients enter the Markov model in fibrosis stages 0 through 4 and after successful therapy move to the SVR stage. Broken arrows indicate a proportional regression of fibrosis. In this model, the regression transition occurs for 5 years after successful treatment. Solid arrows indicate the annual probabilities of liver damage progression after successful treatment.

them by estimating the risk of developing cirrhosis (S1 Materials). We determined the values of these parameters by adjusting them to fit the rate of the progression from chronic hepatitis to liver cirrhosis with a cohort in a published Japanese study [20]. From the adjustment, we estimated that the annual fibrosis progression rates from F0 to F1, F1 to F2, F2 to F3, and F3 to F4 were 0.031, 0.046, 0.071, and 0.068, respectively, in the multivariate model; while those in the random-effects model were 0.077, 0.074, 0.089, and 0.088, respectively.

We obtained other probabilities, such as that from F4 to decompensated cirrhosis, development of HCC at each fibrosis stage and decompensated cirrhosis, mortality rates from decompensated cirrhosis, hepatocellular carcinoma and liver transplantation from the Japanese observational studies [21–24]. We assumed that the mortalities at stage F0 to F4 were the same as those of the age and gender-specific general population, which were obtained from the life-table [25]. Also, the probabilities of receiving a liver transplantation from decompensated LC or HCC were assumed using the data from the report of the liver transplantation registry of Japan [26] (Table 1).

**Table 1. Model parameters (probability).**

| Source State | Target State | Base case | Lower limit | Upper limit | Reference |
|---|---|---|---|---|---|
| **Natural History** | | | | | |
| F0 | F1 (Meta-regression model†) | 0.031 | 0.010 | 0.100 | 19,20 |
| | F1 (Random effects model‡) | 0.077 | 0.067 | 0.088 | 19 |
| | HCC | 0.000 | 0.000 | 0.005 | 25 |
| F1 | F2 (Meta-regression model) | 0.046 | 0.023 | 0.092 | 19,20 |
| | F2 (Random effects model) | 0.074 | 0.064 | 0.086 | 19 |
| | HCC | 0.005 | 0.002 | 0.010 | 25 |
| F2 | F3 (Meta-regression model) | 0.071 | 0.036 | 0.142 | 19,20 |
| | F3 (Random effects model) | 0.089 | 0.077 | 0.103 | 19 |
| | HCC | 0.020 | 0.010 | 0.040 | 25 |
| F3 | F4 (Meta-regression model) | 0.068 | 0.034 | 0.136 | 19,20 |
| | F4 (Random effects model) | 0.088 | 0.075 | 0.104 | 19 |
| | HCC | 0.053 | 0.030 | 0.080 | 25 |
| F4 | Decompensated cirrhosis | 0.056 | 0.025 | 0.098 | 26 |
| | HCC | 0.076 | 0.051 | 0.100 | 25 |
| Decompensated cirrhosis | HCC | 0.076 | 0.051 | 0.100 | 25 |
| | Liver transplantation | 0.004 | 0.003 | 0.004 | 22 |
| | Death | 0.151 | 0.065 | 0.264 | 26 |
| HCC Stage I/II | Liver transplantation | 0.004 | 0.003 | 0.004 | 22 |
| | Death | 0.118 | 0.114 | 0.122 | 26 |
| HCC Stage III/IV | Liver transplantation | 0.004 | 0.003 | 0.005 | 22 |
| | Death | 0.222 | 0.216 | 0.228 | 26 |
| Liver transplantation | Death (First year) | 0.188 | 0.169 | 0.209 | 22 |
| | Death (Succeeding years) | 0.018 | 0.012 | 0.025 | 22 |
| **Fibrosis Progression Post-SVR** | | | | | |
| F0 | F1 | Reduced by 91.4% of pre-SVR probability as listed above | | | 24 |
| F1 | F2 | | | | 24 |
| F2 | F3 | | | | 24 |
| F3 | F4 | | | | 24 |
| **Fibrosis Regression Post-SVR (Only for 5 years after acquiring SVR)** | | | | | |
| F1 | F0 | 0.083 | | | 24 |
| F2 | F1 | 0.159 | | | 24 |
| F3 | F2 | 0.116 | | | 24 |
| F4 | F3 | 0.048 | | | 24 |
| **Hazard ratio of hepatocellular carcinoma from SVR** | | | | | |
| F0,F1,F2,F3 | HCC | 0.240 | 0.120 | 0.360 | 27 |
| F4 | HCC | 0.230 | 0.120 | 0.350 | 27 |

HCC: Hepatocellular carcinoma; F4: Compensated cirrhosis; SVR: sustained virologic response.

† The rates estimated by the multivariate algorithm developed from the results of the meta-analysis study by Thein HH et al. [10] and these parameters by adjusting to fit the rate of the progression from chronic hepatitis to liver cirrhosis with a cohort from a published Japanese study [11].

‡ The progression rates reported for the patients with infection periods of 20 years or more in the meta-analysis study by Thein HH et al. [10, 13, 48, 49].

We validated this natural history model by comparing its predicted survival rate from the model with those obtained in a Japanese observational cohort study of patients with chronic hepatitis [27] (S1 Materials).

### Progression and regression of the fibrosis after SVR

According to previous studies, we assumed that there would be a lower probability of fibrosis progression in the patients achieving an SVR, and that these patients would develop HCC with a lower incidence rate [28]. We also assumed that the liver fibrosis stage in some of the patients at stages F2, F3, or F4 would regress for five years following the achievement of the SVR [29].

### Treatment strategies

We compared the following five strategies in regard to the timing of initiation of DAA treatments, along with a no-treatment strategy (NoRx) (Fig 2).

1) TA: Treat all patients irrespective of their fibrosis stages.

2) F1S: Treat patients with fibrosis stages from F1 to F4 and withhold treatment for patients with stage F0 fibrosis, starting it when fibrosis progresses to stage F1.

3–5) F2S~F4S: Similar to the F1S strategy above, treat patients with F2 to F4 fibrosis (F2S), F3 to F4 fibrosis (F3S), or F4 fibrosis (F4S) and withhold DAA treatment for those with the earlier fibrosis stages.

### DAAs and their effectiveness

Three DAAs were evaluated: sofosbuvir-ledipasvir (SL), glecaprevir-pibrentasvir (GP), and elbasvir (EBV) plus grazoprevir (GZR) (E/G). The duration of treatment was 12 weeks except for patients with non-cirrhosis CH receiving GP, in which it lasted 8 weeks based on the

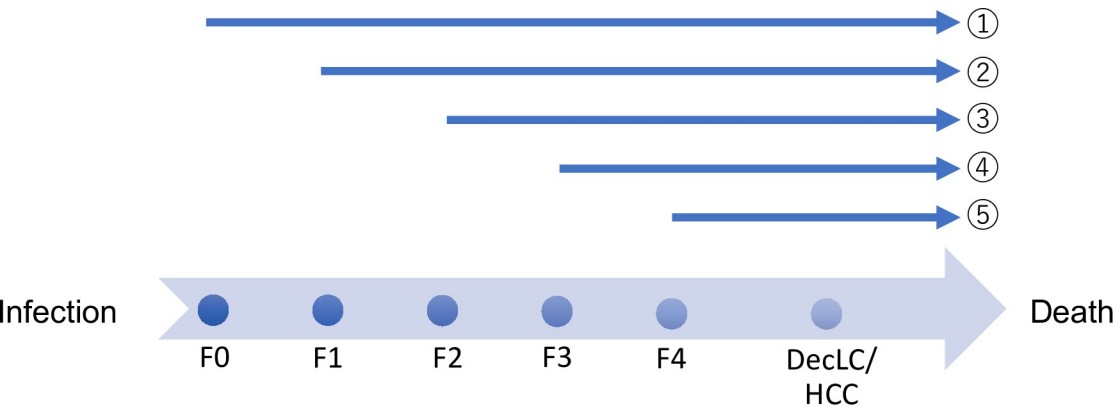

**Fig 2. Treatment strategies.** From Infection to Death, an arrow is shown for HCV natural history, and a dot is shown reflecting the progression of fibrosis stages as well as decompensated liver cirrhosis and hepatocellular carcinoma. Comparisons were made with 6 strategies comprising of 5 timed treatment strategies and a no treatment strategy.

Japanese guidelines [30]. We obtained the rates of achieving SVR and also the rates of discontinuation due to adverse effects from the phase 3 trials conducted in Japan [31–33] (Table 2).

## Disease and treatment costs

We adopted the perspective of health insurance payers and included only direct medical costs per patient. The annual cost associated with each health state was obtained from a Japanese study [34], and costs in the first year and the succeeding years of liver transplantation were obtained from a Japanese cost study of liver transplantation [21] and adjusted according to purchasing power parities.

In particular, we assumed that the annual costs of F0 and F1 stage treatments were the same as those of treating inactive chronic hepatitis, and the costs of F2 and F3 stage treatments were the same as those of treating active chronic hepatitis.

Costs for the three DAAs were obtained from the 2018 National Health Insurance drug price lists. We calculated the total costs of DAA treatments by multiplying the price of each DAA by the daily dose and the duration of regular treatment (days). We did not consider the costs caused by adverse effects of DAAs, because they were rarely serious and because we could not obtain reliable data for the costs of treating adverse effects. We estimated the monitoring cost at pretreatment, during treatment and post-treatment by the micro-costing of reimbursements of the National Health Insurance system for their standard model of regular office visits. We expressed all costs converted from Japanese yen to the US dollar, and the currency exchange rate was 110 Japanese yen per US dollar (Table 3).

## Health-related quality of life

We applied the utility values of each health status from the results of a survey given to Japanese patients and experienced hepatologists [35]. Though the increments of the utility value after SVR achieved by DAA treatment differed from study to study [36–39], we adopted the results of a study on Japanese patients using SF-6D [39], and we assumed the increment was 0.022 in the base analysis (Table 4).

**Table 2. Baseline characteristics and main outcomes of the phase 3 studies of DAAs.**

|  | sofosbuvir-ledipasvir | | glecaprevir-pibrentasvir | | elbasvir-grazoprevir | |
|  | Mizokami et al. 2015[31] | | Chayama et al. 2018[32] | | Kumada et al. 2017[33] | |
|  | **CH** | **LC** | **CH** | **LC** | **CH** | **LC** |
|---|---|---|---|---|---|---|
| **Treatment-naive patient** | N = 70 | N = 13 | N = 129 | N = 38 | N = 227 | N = 35 |
| **Baseline Characteristics** |  |  |  |  |  |  |
| Age, mean±SD, years | 60±9.2† | | | | 61±12.5 | 64.8±9.2 |
| Age, median (range), years | | | 64 (21–86) | 73 (48–85) | | |
| Male, n (%) | 69 (40)† | | 47 (36) | 17 (45) | 87 (38) | 18 (51) |
| **Virological response (SVR12 or SVR24)** |  |  |  |  |  |  |
| n (%) | 70 (100) | 13 (100) | 128 (99) | 38 (100) | 219 (96) | 34 (97) |
| CI, range, (%) | 95–100 | 75–100 | 98–100 | 91–100 | 95–98 | 94–100 |
| **Discontinuation by side effect** |  |  |  |  |  |  |
| n (%) | 0 (0) | 0 (0) | 0 (0) | 1 (3) | 3 (1) | 0 (0) |

CH: chronic hepatitis; LC: compensated cirrhosis; SVR: sustained virologic response.

†N = 171: Due to the lack of separated data on the age and male ratio of treatment-naïve patients, these values were the average of 171, patients including the previously treated patients.

**Table 3. Model parameters (cost).**

| Variable (US dollars) | | | Base case | Lower limit | Upper limit | Reference |
|---|---|---|---|---|---|---|
| **Annual healthcare costs by disease state** | | | | | | |
| | Chronic hepatitis | | | | | |
| | | F0, F1 state | 1,110 | 550 | 1,660 | 26 |
| | | F2, F3 state | 3,140 | 1,570 | 4,700 | 26 |
| | | SVR state | 240 | 130 | 370 | † |
| | Compensated cirrhosis | | 4,350 | 2,180 | 6,530 | 26 |
| | | SVR state | 490 | 240 | 720 | † |
| | Decompensated cirrhosis | | 6,420 | 3,210 | 9,640 | 26 |
| | Hepatocellular carcinoma | | | | | |
| | | Stage I/II | 10,440 | 5,220 | 15,670 | 26 |
| | | Stage III/IV | 18,120 | 9,060 | 27,180 | 26 |
| | Liver Transplantation | | | | | |
| | | First year | 129,090 | 64,550 | 193,640 | 24 |
| | | Succeeding years | 17,380 | 8,690 | 26,070 | 24 |
| **Cost of treatment modality** | | | | | | |
| | sofosbuvir—ledipasvir | | | | | |
| | | Daily SL cost | 500 | 250 | 750 | ¶ |
| | | Total drug costs (12 weeks) | 41,840 | 20,920 | 62,770 | ¶ |
| | | Treatment-related medical care costs ‡ | 740 | | | § |
| | glecaprevir—pibrentasvir | | | | | |
| | | Daily GP cost | 660 | 330 | 990 | ¶ |
| | | CH | | | | |
| | | | Total drug costs (8 weeks) | 36,980 | 18,490 | 55,470 | ¶ |
| | | | Treatment-related medical care costs ‡ | 530 | | | § |
| | | LC | | | | |
| | | | Total drug costs (12 weeks) | 55,460 | 27,730 | 83,190 | ¶ |
| | | | Treatment-related medical care costs ‡ | 740 | | | § |
| | elbasvir—grazoprevir | | | | | |
| | | Daily Elbasvir cost | 245 | 123 | 369 | ¶ |
| | | Daily Grazoprevir cost | 175 | 88 | 263 | ¶ |
| | | Total drug costs (12 weeks) | 35,220 | 17,610 | 52,830 | ¶ |
| | | Treatment-related medical care costs ‡ | 740 | | | § |

†Assumption

‡Office visit and laboratory test

§Expert consensus

¶The 2016 National Health Insurance drug price list

## Model assumptions

We made several assumptions for the models as follows.

1. The patients who failed DAA treatment did not under undergo additional alternative DAA treatments and were at risk of natural CHC progression and related complications.

2. The utilities among the patients with fibrosis stages between F0 to F3 were the same value because the data needed to differentiate the utility values among the fibrosis stages is unavailable in Japan.

**Table 4. Model parameters (quality of life weights).**

| Health State | | Base case | Lower limit | Upper limit | Reference |
|---|---|---|---|---|---|
| Chronic hepatitis | | 0.821 | 0.780 | 0.850 | 26 |
| | SVR state | 0.843 † | 0.832 | 0.854 | 39 |
| Compensated cirrhosis | | 0.737 | 0.680 | 0.790 | 26 |
| | SVR state | 0.759 † | 0.748 | 0.770 | 39 |
| Decompensated cirrhosis | | 0.671 | 0.610 | 0.730 | 26 |
| Hepatocellular carcinoma | | | | | |
| | Stage I/II | 0.675 | 0.620 | 0.730 | 26 |
| | Stage III/IV | 0.428 | 0.370 | 0.490 | 26 |
| Liver Transplantation | | | | | |
| | First year | 0.651 | 0.590 | 0.700 | 26 |
| | Succeeding years | 0.651 | 0.590 | 0.700 | 26 |

†The utility increment after SVR achieved by DAA treatment is 0.022.

3. We did not consider DAA treatment for patients with decompensated cirrhosis.

4. Liver transplantation is performed only for patients with decompensated cirrhosis or HCC.

5. Cost and utility loss from adverse effects were not considered, as they did not seem to be significantly different between the treatment strategies except for that of no-treatment.

6. We did not consider analyzing mutations, as they were not covered by the national insurance system and performed routinely in current practice for treatment-naïve patients.

## Model simulation

We performed the model simulation at a cycle of 1 year under the lifetime horizon. We also discounted both the costs and outcome at 2% per year according to Japanese guidelines [40].

## Model population

The characteristics of patients in the cohort for the analysis of each treatment strategy were specified based on data from a phase 3 DAA treatment study for Japanese patients with genotype 1 CHC [41]. The mean age of the subjects was 57 years old, 44.3% were male, and the baseline distribution of patients across fibrosis stages was as follows: F0, 36.7%; F1, 20.9%; F2, 16.3%; F3, 17.2%; and F4, 8.8%. The age of the cohort and the rates of fibrosis stages in the cohort were subjected to a sensitivity analysis (SA).

## Model outcomes

The model estimated lifelong costs gained per patient and life years (LY), and the gained quality-adjusted life years (QALY). The incremental cost-effectiveness ratio (ICER) of TA relative to the other treatment timing strategies was also estimated. The cost-effectiveness threshold (willingness-to-pay) of the ICER relative to the next most cost-effective strategy was assumed to be US$50,000 per QALY.

## Sensitivity Analysis (SA)

To clarify the uncertainty around model input parameters affecting the ICER, we conducted both a deterministic sensitivity analysis (DSA) and a probabilistic sensitivity analysis (PSA).

In the DSA, we performed tornado analyses as a one-way SA in which the distribution of the fibrosis stages among the model population, transitional probabilities, mortality rates by decompensated cirrhosis and hepatocellular carcinoma, annual health state cost, health-related utility weights, and discounting rates were varied simultaneously to show the impact of each variable on the model results. Ranges of the variables for the DSA were set by 95% confidential intervals from the primary data resources, if such data were available or set from 50% to 150% of the base case values, if such data were not available. The range of each variable is shown in Table 1.

In the PSA, each input parameter was assumed to be associated with a certain probabilistic distribution. Transition probabilities, mortality rates, SVR rates (except for that of SL, which was assumed to follow the uniform distribution), and QOL weights were assumed to follow the beta distribution. We also assumed that the health costs followed the gamma distribution. The results are shown in the multiple cost-effectiveness acceptability curves (CEACs), where the horizontal axis indicates the willingness-to-pay (WTP) threshold for one additional QALY with a range of US$0 to US$100,000, and the vertical axis indicates the probability of each treatment being the most effective. The range of each variable for the PSA is shown in Table 5.

## Results

### Base-case analysis

We showed mainly the base case results simulated with the fibrosis progression rates by the meta-regression model, since they were the slower progression rates and were more conservative figures compared with those from the random-effects model, which are presented in S1 Materials.

The model predicted that the patients without any treatment would suffer from decompensated cirrhosis at 10% probability, hepatocellular carcinoma at 38% probability, and die from a liver-related cause at 34% probability. On the other hand, it also predicted that TA with DAAs would prevent the progression to decompensated cirrhosis, hepatocellular carcinoma, and liver-related death at probabilities of 0%, 22.1% and 17.7% with SL; 0.5%, 22.8% and 18.4% with GP; and 3.3%, 25.0% and 20.9% with E/G, respectively (Fig 3).

Fig 4 shows the base results in the cost-effectiveness plane. TA was the most effective strategy in QALY, followed by F1S, F2S, F3S, and F4S, irrespective of difference in DAAs. TA yielded gains of 0.12, 0.12, and 0.11 quality-adjusted life years (QALY) compared with F1S and gains of 0.33, 0.33, and 0.32 QALY compared with F2S by the treatment with SL, GP, and E/G, respectively. Regarding the lifetime cost of the strategies, F2S had the lowest cost, followed by F1S and TA. Therefore, these strategies were preferable to F3S and F4S as well as no treatment. As a result, the incremental cost-effectiveness ratio (ICER) per QALY gains of TA against F1S and those of F1 against F2S were $39,780 and $15,960 by SL, $30,260 and $11,530 by GP, and $29,490 and $11,050 by E/G, respectively. Thus, the ICERs of TA with all DAAs were lower than those of the thresholds of cost-effectiveness compared with the other strategies (Table 6). Though we can differentiate between the stages of F0 (no fibrosis) and F1(mild fibrosis) by histological examination, this distinction is not considered worthwhile in a clinical setting, because the clinician is not likely to use histological examination and cannot easily distinguish these states without it. Therefore, we combined these stages as F0/F1 and demonstrated the cost-effectiveness between TA and F2S as follows.

### DSA

We showed the results of the tornado diagrams which presented the effect of variation in key model variables on the ICER of TA against F2S of each input in the model of three DAA

**Table 5. PSA variable range.**

| Model parameters | Type | Range 2.5% | Range 97.5% |
|---|---|---|---|
| **Treatment efficacy** | | | |
| sofosbuvir-ledipasvir | | | |
| CH stage | Triangular | 0.966 | 1.000 |
| LC stage | Triangular | 0.807 | 0.997 |
| glecaprevir-pibrentasvir | | | |
| CH stage | Beta | 0.972 | 1.000 |
| LC stage | Triangular | 0.933 | 0.999 |
| elbasvir-grazoprevir | | | |
| CH stage | Beta | 0.937 | 0.984 |
| LC stage | Beta | 0.897 | 0.999 |
| **Transition probabilities** | | | |
| F0 to F1 | | | |
| Meta-regression model | Simulation† | 0.021 | 0.044 |
| Random effect model | Beta | 0.104 | 0.130 |
| F1 to F2 | | | |
| Meta-regression model | Simulation† | 0.030 | 0.056 |
| Random effect model | Beta | 0.075 | 0.095 |
| F2 to F3 | | | |
| Meta-regression model | Simulation† | 0.045 | 0.097 |
| Random effect model | Beta | 0.109 | 0.131 |
| F3 to LC | | | |
| Meta-regression model | Simulation† | 0.051 | 0.091 |
| Random effect model | Beta | 0.105 | 0.128 |
| LC to decLC | Beta | 0.025 | 0.098 |
| F1 to HCC | Beta | 0.001 | 0.011 |
| F2 to HCC | Beta | 0.010 | 0.033 |
| F3 to HCC | Beta | 0.029 | 0.085 |
| LC, decLC to HCC | Beta | 0.055 | 0.107 |
| Proportion of Stage I/II | Beta | 0.877 | 0.954 |
| Death by decLC | Beta | 0.066 | 0.259 |
| Death by HCC Stage I/II | Beta | 0.114 | 0.122 |
| Death by HCC Stage III/IV | Beta | 0.216 | 0.228 |
| Death by LT (First year) | Beta | 0.169 | 0.208 |
| Death by LT (Succeeding years) | Beta | 0.012 | 0.025 |

| Model parameters | Type | Range 2.5% | Range 97.5% |
|---|---|---|---|
| **Distribution of fibrosis states** | | | |
| F0 | Dirichlet | 0.303 | 0.434 |
| F1 | Dirichlet | 0.157 | 0.267 |
| F2 | Dirichlet | 0.116 | 0.214 |
| F3 | Dirichlet | 0.125 | 0.226 |
| F4 | Dirichlet | 0.055 | 0.129 |
| **Annual healthcare costs by disease state (US dollars)** | | | |
| Chronic hepatitis | | | |
| F1, F2 state | Gamma | 100 | 3,400 |
| F3 state | Gamma | 0 | 13,200 |
| SVR state | Gamma | 200 | 300 |
| Compensated cirrhosis | Gamma | 100 | 17,700 |
| SVR state | Gamma | 400 | 600 |
| Decompensated cirrhosis | Gamma | 500 | 19,700 |
| Hepatocellular carcinoma | | | |
| Stage I/II | Gamma | 1,400 | 28,800 |
| Stage III/IV | Gamma | 2,200 | 49,900 |
| Liver Transplantation | | | |
| First year | Gamma | 99,100 | 162,900 |
| Succeeding years | Gamma | 13,300 | 22,000 |
| **QOL weight** | | | |
| Chronic hepatitis | Normal | 0.757 | 0.876 |
| Compensated cirrhosis | Normal | 0.677 | 0.794 |
| Decompensated cirrhosis | Normal | 0.611 | 0.728 |
| Hepatocellular carcinoma | | | |
| Stage I/II | Normal | 0.615 | 0.732 |
| Stage III/IV | Normal | 0.369 | 0.487 |
| Liver Transplantation | Normal | 0.645 | 0.657 |
| Utility increments after SVR | Normal | 0.013 | 0.034 |

†The range of the meta-regression model is the simulation result with parameters for each distribution.

treatments in Fig 5. The only variable which significantly influenced the ICER so that it exceeded the threshold of $50,000/QALY was the age of the cohort. The age at commencement of DAA therapy greatly affected the ICER of TA against F2S. If the age was less than 40 years, the lifetime cost of TA was less than F2S, and TA became a cost-saving strategy. If the age exceeded 65 years old in the comparison between TA and F2S, each ICER was estimated to be higher than the upper limit of the CEA, $50,000/QALY, respectively. The different DAAs were associated with different age thresholds—namely, 65, 68, and 68 years for SL, GP, and E/G, respectively. Therefore, we ought to lower the base price of DAAs by 59% in the case of SL and 67% in the cases of GP and E/G in order for the ICER of TA against F2S for the cohort of patients aged 75 years to become cost-effective.

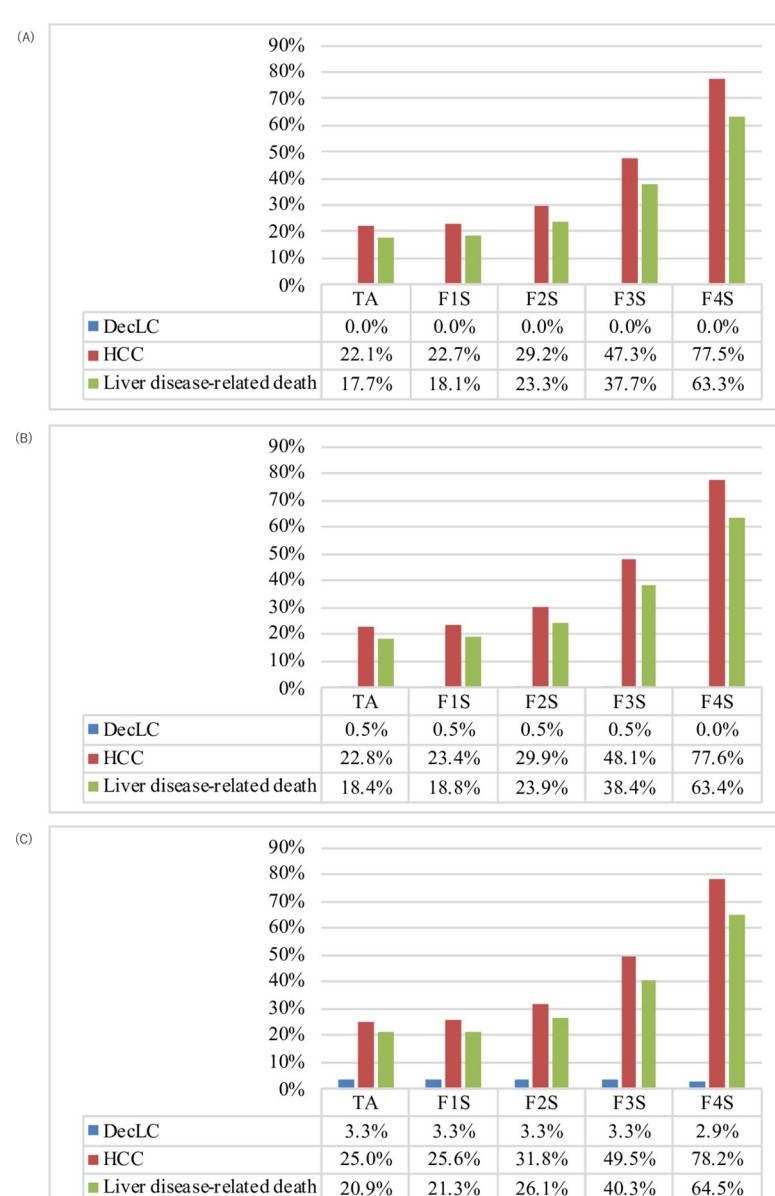

**Fig 3. Estimated relative risk of liver disease in the meta-regression model.** Dec LC: decompensated cirrhosis; HCC: hepatocellular carcinoma; "Liver disease-related Death" represents the disease-specific mortality associated with having decompensated cirrhosis, liver transplant, or hepatocellular carcinoma. The degree of disease reduction in the five treatment strategies compared to the no-treatment strategy, with SL (A), GP (B), and E/G (C).

## PSA

According to the results of the second-order Monte Carlo simulation presented in Fig 6, TA was found to be cost-effective in at least 51.6% of the 10000 PSA iterations run, with a WTP threshold of up to $50,000 per QALY, regardless of the method used to estimate the progression rates of fibrosis. Among the three DAA options, the acceptable probability of strategies of starting treatment at stage F2 or higher was less than 4% for the progression rates estimated by the meta-regression model and less than 0.5% for those estimated by the random-effects model.

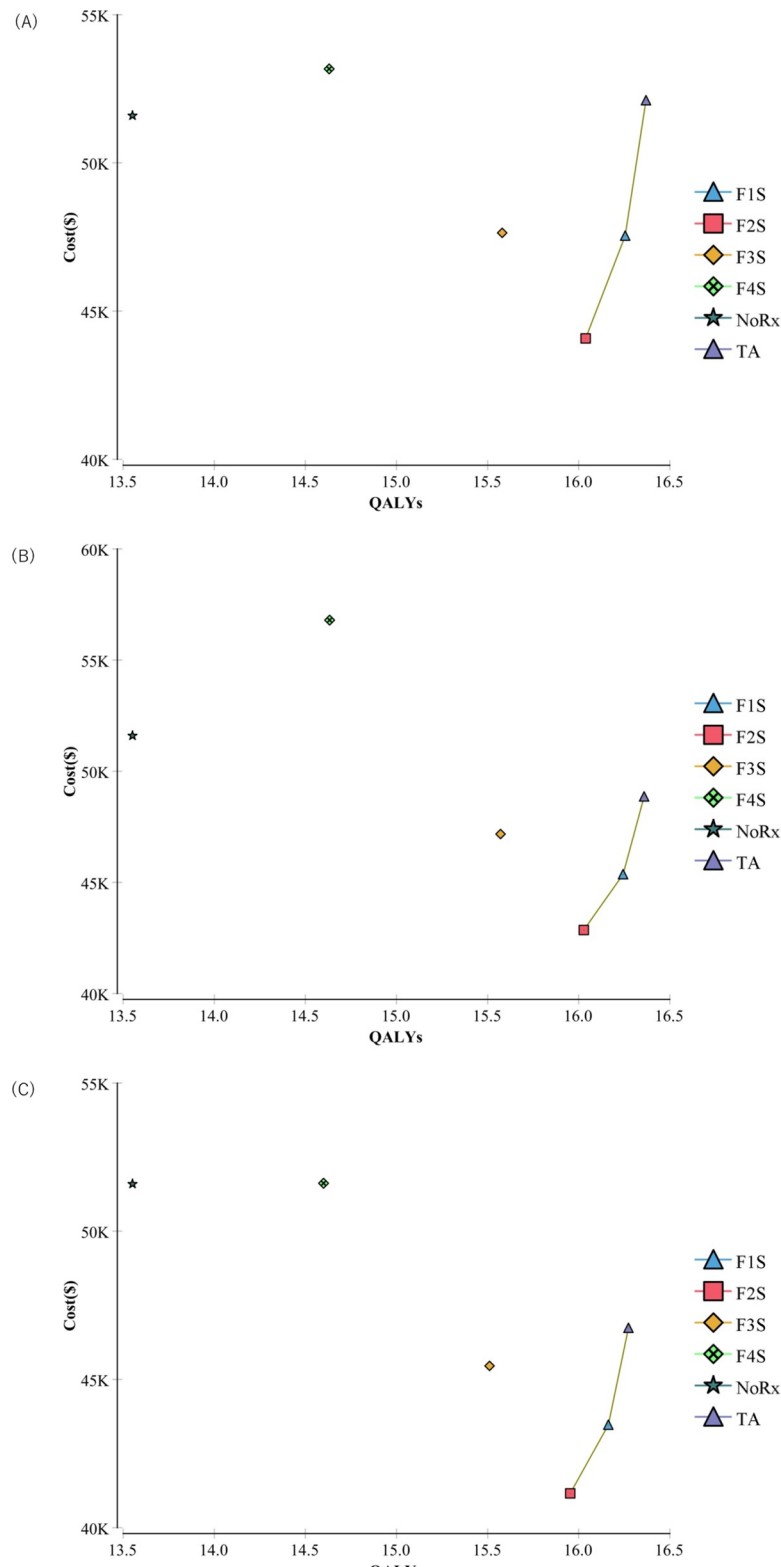

**Fig 4. Base case analysis in the meta-regression model.** Base case analysis performed in the meta-regression model of treatment with (A) SL, (B) GP, and (C) E/G.

**Table 6. Base case results of HCV Treatment.**

| Strategy | Lifetime Costs ($) | Δcost ($) | LY | ΔLY | QALYs | ΔQALY | ICER /LY ($) | ICER /QALY ($) |
|---|---|---|---|---|---|---|---|---|
| Sofosbuvir—Ledipasvir | | | | | | | | |
| Meta-regression model | | | | | | | | |
| F2S | 44,080 | | 19.52 | | 16.04 | | Ref | Ref |
| F1S | 47,530 | 3,450 | 19.66 | 0.14 | 16.25 | 0.21 | 24,500 | 15,960 |
| TA | 52,100 | 4,570 | 19.67 | 0.01 | 16.37 | 0.12 | 552,020 | 39,780 |
| TA vs. F2S | | 8,020 | | 0.15 | | 0.33 | 53,470 | 24,300 |
| Random effects model | | | | | | | | |
| F2S | 48,210 | | 19.48 | | 16.03 | | Ref | Ref |
| F1S | 50,520 | 2,310 | 19.65 | 0.17 | 16.27 | 0.24 | 13,800 | 9,430 |
| TA | 52,180 | 1,660 | 19.66 | 0.01 | 16.36 | 0.09 | 99,100 | 19,090 |
| TA vs. F2S | | 3,970 | | 0.18 | | 0.33 | 22,060 | 12,030 |
| Glecaprevir—Pibrentasvir | | | | | | | | |
| Meta-regression model | | | | | | | | |
| F2S | 42,860 | | 19.51 | | 16.03 | | Ref | Ref |
| F1S | 45,350 | 2,490 | 19.65 | 0.14 | 16.24 | 0.21 | 17,820 | 11,530 |
| TA | 48,850 | 3,500 | 19.66 | 0.01 | 16.36 | 0.12 | 425,910 | 30,260 |
| TA vs. F2S | | 5,990 | | 0.15 | | 0.33 | 39,930 | 18,150 |
| Random effects model | | | | | | | | |
| F2S | 46,450 | | 19.46 | | 16.01 | | Ref | Ref |
| F1S | 47,880 | 1,430 | 19.63 | 0.17 | 16.26 | 0.25 | 8,600 | 5,850 |
| TA | 48,980 | 1,100 | 19.64 | 0.01 | 16.34 | 0.08 | 66,000 | 12,590 |
| TA vs. F2S | | 2,530 | | 0.18 | | 0.33 | 14,060 | 7,670 |
| Elbasvir—Grazoprevir | | | | | | | | |
| Meta-regression model | | | | | | | | |
| F2S | 41,160 | | 19.44 | | 15.95 | | Ref | Ref |
| F1S | 43,460 | 2,300 | 19.57 | 0.13 | 16.16 | 0.21 | 16,950 | 11,050 |
| TA | 46,730 | 3,270 | 19.58 | 0.01 | 16.27 | 0.11 | 408,810 | 29,490 |
| TA vs. F2S | | 5,570 | | 0.14 | | 0.32 | 39,790 | 17,410 |
| Random effects model | | | | | | | | |
| F2S | 44,720 | | 19.38 | | 15.93 | | Ref | Ref |
| F1S | 46,000 | 1,280 | 19.54 | 0.16 | 16.16 | 0.23 | 7,900 | 5,400 |
| TA | 46,990 | 990 | 19.56 | 0.02 | 16.25 | 0.09 | 61,290 | 11,820 |
| TA vs. F2S | | 2,270 | | 0.18 | | 0.32 | 12,610 | 7,090 |

LY: Life years; QALY: Quality-adjusted life years; ICER: Incremental cost-effectiveness ratio; Ref: Reference.

† F3S, F4S, NoRx: Dominated strategies by F2S.

## Discussion

The current DAAs can achieve a high SVR rate of more than 95% among patients with hepatitis C genotype 1 and their adverse effects are much less harmful compared with those of previous treatments, including interferon or ribavirin [42–46]. The evidence of reduction of the HCV-related mortality by DAA treatment in those patients without advanced fibrosis had been lacking, although the peginterferon-based treatment was known to reduce the all-cause mortality [47, 48]. However, a recent study revealed that DAAs reduced the mortality of such patients [49]. Moreover, the WHO published a review of studies evaluating the cost-

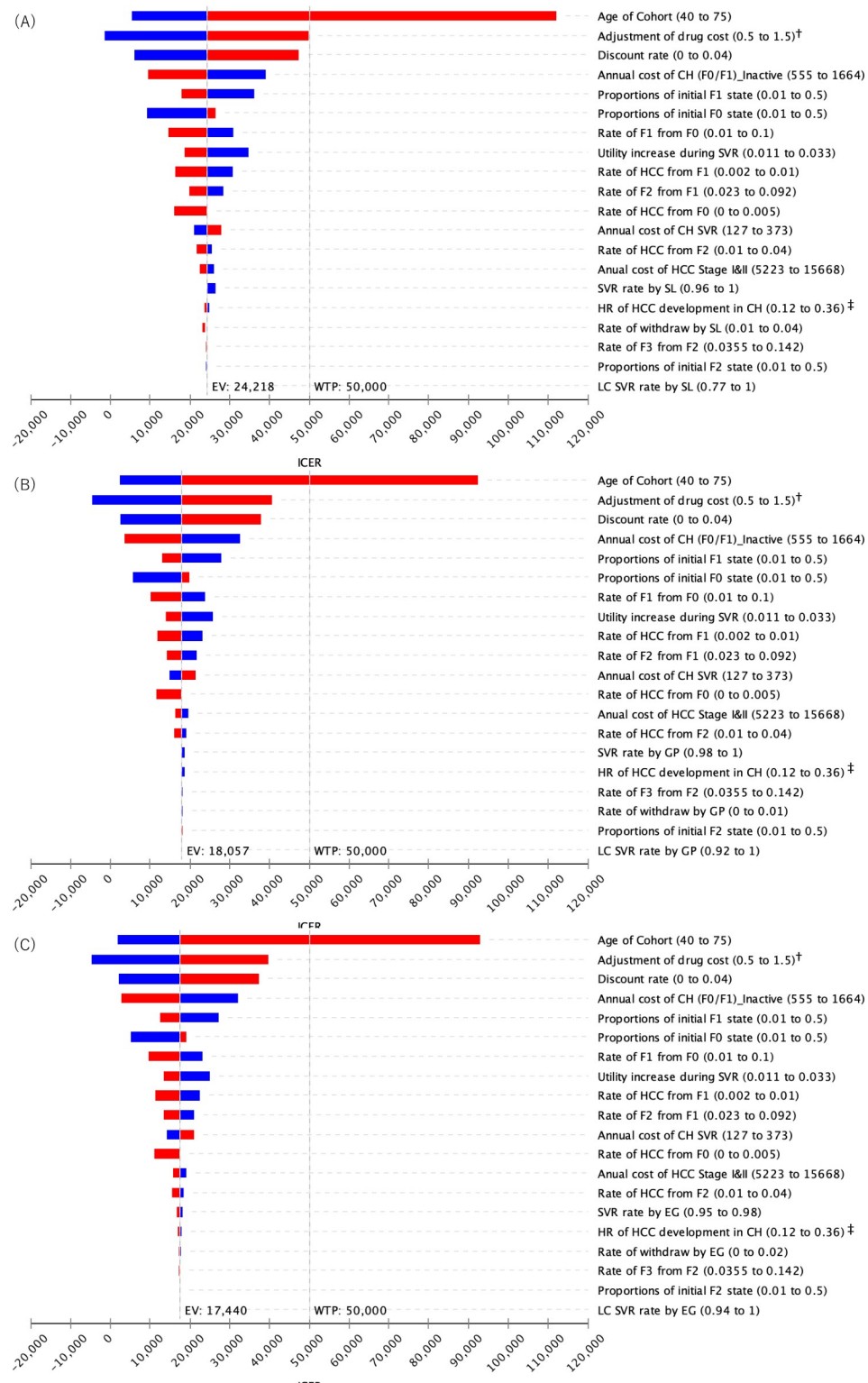

**Fig 5. Tornado diagrams for TA vs. F2S.** †The adjustment range of drug costs was 0.5 to 1.5 times the basic cost. ‡ Hazard ratio of HCC development with SVR compared to non-SVR in CH. Tornado diagrams performed by meta-regression model of treatment with (A) SL, (B) GP, and (C) E/G. The result was sensitive to the age of the cohort applied to costs and outcomes.

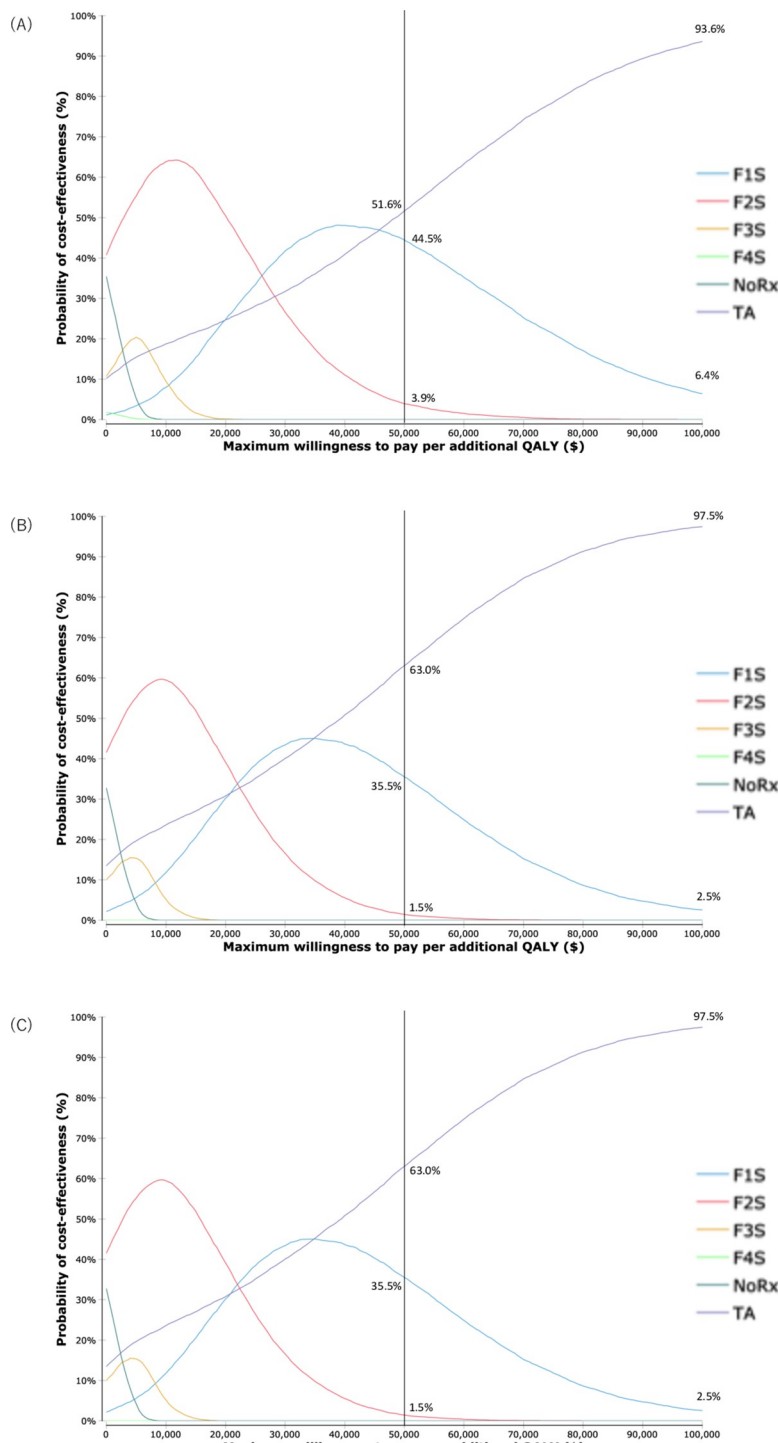

**Fig 6. Cost-effectiveness acceptability curves of meta-regression model.** Cost-effectiveness acceptability curves performed by meta-regression model of treatment with (A) SL, (B) GP, and (C) E/G. When the WTP threshold is set at $50,000 per QALY, the probability of TA being cost-effective was 51.6%, 63.0%, and 63.0% by the treatment with SL, GP, and E/G, respectively.

effectiveness of "Treat All" policy for the patients with HCV, and revealed its feasibility and cost-effectiveness in many countries [14].

Our present study similarly showed the cost-effectiveness of DAA treatments for Japanese patients with CHC genotype 1 under conditions of the circumstance in Japan with only one exception: patients with no fibrosis or only mild fibrosis who are older than 65 years.

Although we presented the results of the three DAAs which are currently available in Japan and found that they had slightly different prices and effectiveness levels, there was not so much difference in cost-effectiveness among the strategies used to treat patients without consideration of their fibrotic stages. These results were consistent with other previous cost-effectiveness studies from other countries on the treatment of early stage of fibrosis for the patients with CHC genotype 1 [14, 28, 50] and should support the policy of the current Japanese guideline recommendations that all Japanese HCV patients of genotype 1 should be eligible for the DAA treatment [30].

Thus, the results suggested that if there is no contra-indication, the commencement of DAA treatment for the patients with any fibrosis stage would be more cost-effective than waiting for the progression to an advanced fibrosis state defined as F2 or higher in the guidelines.

Therefore, we no longer have to know patient's exact fibrotic stage at the time of DAA treatment commencement from the viewpoint of cost-effectiveness, but the knowledge of patients' fibrosis stages of is still important for predicting the likelihood of HCC development even after they achieved the SVR state with DAA treatments.

Liver biopsy is the gold standard method for accurate confirmation of the stages of liver fibrosis, but it is also an invasive procedure that carries risks of severe complications such as bleeding and tumor dissemination. Nowadays, we can roughly estimate it by laboratory findings such as platelet counts, APRI score, and Fib-4 score, or by the US-based or MRI elastography [51].

The cohort age at commencement of DAA treatment is a critical factor affecting the ICER. The CEA results showed that it is preferable to initiate DAA treatment at a younger age. Under the meta-regression model of fibrosis stage progression and the treatment of SL, patients younger than 65 years were eligible for the DAA treatment. However, Asahina Y et al. reported accelerating fibrosis progression rates among their elderly patients in Japan, and they noted that the rate of progression of fibrosis over time was 0.21±0.10 fibrosis stages per year in older patients aged 65 years or more, and 0.03±0.21 fibrosis stages per year in the younger patients group [52]. When we applied this higher transition probability for simulated patients of aged 65 years and older, the upper limits of the age of the cohort for which the ICER between TA and F2S fell in the range of cost-effectiveness became 75, 76, and 77 years old, respectively.

Although we set the degree of gain in the utilities of the SVR state from the non-SVR state as 0.022 in the base case analysis, we cannot rule out the possibility that this value affected the results. So far, the relevant evidence regarding the degree of the gain by SVR yielded by the DAAs has been limited. One recent study showed no significant improvement in quality of life during or after treatment [36], while three studies showed improvement in quality of life compared to those of the prior treatments [37–39, 53]. Most of the other cost-effectiveness analyses related to antiviral treatment for chronic hepatitis C used a utility value equal to or higher than 0.05 for the SVR state [53–57]. In this study, we applied the values of Younossi ZM et al., who found that the SVR gained 0.022 in the utility score from the Japanese population after treatment with ledipasvir and sofosbuvir, using SF-6D measurement for the QOL improvement in the base case analysis [39]. This survey was a unique one in that the health utilities of the SVR state were obtained among Japanese HCV-infected patients. Therefore, we performed a sensitivity analysis varying the utility score from 0.011 to 0.033 and showed that there were no cases in which the ICER exceeded the threshold of WTP of US$50,000.

## Limitations

Our study had several limitations. First, the generalizability of the results was limited to Japanese patients because our model was constructed using transition probability obtained mainly from Japanese observational studies, and we only considered a direct cost based on the Japanese national insurance system. Notably, we used the progression rates of fibrosis estimated by the meta-regression model of Thein et al. [19] with parameters adjustment within the appropriate ranges according to Japanese epidemiological studies. Although we validated the model using the rates of development of liver cirrhosis and mortality rates from a Japanese cohort study, the estimated progression rates were slower than those of other studies. Moreover, we applied a slightly higher probability for the development of hepatocellular carcinoma in patients with compensated or decompensated liver cirrhosis, which we also obtained from a Japanese epidemiological study [22]. This higher probability should cause higher mortality and costs when treatment is initiated at a more advanced fibrosis stage than at a mild fibrosis stage or in patients with no fibrosis, and the sensitivity analysis confirmed that the effects of early initiation of DAA treatments were more favorable. However, even when we applied a sufficiently low value, the ICER between TA and F1S did not exceed the threshold.

Second, we did not consider separately the age and sex-adjusted mortality rate related to hepatitis C virus infection in our model due to the lack of relevant data, and we applied the general mortality rate instead of the hepatitis-specific mortality.

Third, we could not obtain robust evidence of the transitional probability and cost estimation of liver transplantation. In Japan, however, the annual number of liver transplantations due to various causes, including cases with decompensated liver cirrhosis and HCC, is about 500 [28], and this number corresponds to several percents of the total number of patients with hepatocellular carcinoma. Therefore, this uncertainty was considered to have little effect on the results.

Fourth, we did not consider the effect on extrahepatic morbidity and avoidance of HCV infection. If we were to include these effects in our model, the cost-effectiveness of TA would almost certainly improve.

Although we need more precise models to overcome these limitations in future studies, our results showed the current estimation of the cost-effectiveness of the initiation of DAA treatment regardless of liver fibrotic stages among the patients with HCV infection genotype 1.

In conclusion, our results suggested that the treatment of all Japanese patients younger than 65 years of age with genotype 1 CHC, irrespective of their liver fibrosis stage, would be cost-effective.

## Supporting information

**S1 Materials.**
(DOCX)

## Author Contributions

**Conceptualization:** Riichiro Suenaga, Haku Ishida.

**Data curation:** Machi Suka, Tomohiro Hirao, Haku Ishida.

**Formal analysis:** Riichiro Suenaga, Haku Ishida.

**Funding acquisition:** Tomohiro Hirao.

**Supervision:** Isao Hidaka, Isao Sakaida.

**Validation:** Machi Suka, Tomohiro Hirao, Haku Ishida.

**Writing – original draft:** Riichiro Suenaga, Haku Ishida.

**Writing – review & editing:** Riichiro Suenaga, Machi Suka, Tomohiro Hirao, Isao Hidaka, Isao Sakaida, Haku Ishida.

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
