## [Decision Letter · Decision Letter 0]

18 Dec 2020

PONE-D-20-30605

Cost-effectiveness of a “treat-all” strategy using direct-acting antivirals (DAAs) for Japanese patients with chronic hepatitis C genotype 1 at different fibrosis stages

PLOS ONE

Dear Dr. Suenaga,

Thank you for submitting your manuscript to PLOS ONE. After careful consideration, we feel that it has merit but does not fully meet PLOS ONE’s publication criteria as it currently stands. Therefore, we invite you to submit a revised version of the manuscript that addresses several points raised during the review process.

We look forward to receiving your revised manuscript.

Kind regards,

Isabelle Chemin, PhD

Academic Editor

PLOS ONE

Journal Requirements:

'I have read the journal's policy and the authors of this manuscript have the following competing interests:

Isao Sakaida received a Lectures fee from Gilead.

Isao Hidaka received a Lectures fee from Gilead and Abbvie.'

a. Please confirm that this does not alter your adherence to all PLOS ONE policies on sharing data and materials, by including the following statement: "This does not alter our adherence to  PLOS ONE policies on sharing data and materials.” (as detailed online in our guide for authors http://journals.plos.org/plosone/s/competing-interests).  If there are restrictions on sharing of data and/or materials, please state these.

Please note that we cannot proceed with consideration of your article until this information has been declared.

Reviewers' comments:

Reviewer's Responses to Questions

**Comments to the Author**

1. Is the manuscript technically sound, and do the data support the conclusions?

Reviewer #1: Yes

2. Has the statistical analysis been performed appropriately and rigorously? 

Reviewer #1: Yes

3. Have the authors made all data underlying the findings in their manuscript fully available?

Reviewer #1: Yes

4. Is the manuscript presented in an intelligible fashion and written in standard English?

Reviewer #1: Yes

5. Review Comments to the Author

Reviewer #1: Since the first direct-acting antivirals (DAAs) received US Food and Drug Administration approval in 2011, several cost-effectiveness investigations have compared DAA-based regimens to previous standard-of-care regimens to calculate ICERs. They have also investigated the cost-effectiveness of eliminating HCV treatment restrictions. Compared to interferon-based regimens, the ICER for DAAs has consistently been estimated at <$100,000 per QALY for all genotypes and fibrosis stages.

Several studies have compared DAA regimens against one another. In general, when given a choice between recommended HCV DAA regimens, the less costly regimen is preferred as a more efficient use of resources (even if it requires multiple tablet dosing). Because of the similar efficacy of most DAA regimens, cost becomes the critical factor driving relative cost-effectiveness. Studies have also estimated the cost-effectiveness of HCV treatment in special populations, including patients awaiting liver transplantation, HIV/HCV-coinfected patients, those with chronic kidney disease, persons who inject drugs, and adolescents—all with favorable ICERs. At this time, it is reasonable to conclude that DAA regimens provide good value for the resources invested. That is the reason why AASLD and EASL guidelines have endorsed a treat all approach.

That being said, the analysis presented here has relevance due to 1) its focus on the Japanese population with GT1 HCV and 2) tweak association between willingness-to-pay and the real-world bottom line. Societal willingness-to-pay thresholds in CEAs are not based on actual budget calculations and have little relationship to a payer’s bottom line. Under these premises, the results of the study presented here are important to give Japanese policy makers estimates of the cost of eliminating HCV. This is a timely study as Japan just this week has released its HCV elimination plan. Please consider adding these elements to the background section. The study objective goes beyond the simple verification of ICER below $50,000 to $100,000, by providing contemporary cost perspective to national elimination plans underway.

Minor revisions:

1. The segment below need a few linguistic corrections.

Fig 2 Treatment strategies

180 Infection to Death arrow is shown HCV natural history. and dot is shown reflecting progression

181 of fibrosis stages as well as decompensated liver cirrhosis and hepatocellular carcinoma.

182 Comparisons were made with 6 strategies that is 5 timing treatment strategy and no treatment

183 strategy.

2. Table 2 in line 191: needs formatting as follows: a) table title needs to be more specific, as “treatment effects” says little about the information covered by the table; b) abbreviations such as CH and LC need to be translated in the bottom of the table (chronic hepatitis? Liver cirrhosis?); c) the footnote statement “†Including the previously treated patients. (N=171)” seems confusing to me, as I do not see 171 patients in the SL trial, only 83 patients. If there is another SL study in Japanese populations, I suggest including this study too so the table can show all 171 patients as stated.

3. Table 3 line 214: in regards to the cost of each treatment regimen, many patients receiving SL can be treated for 8 weeks only (viral load less than 6 million, non-black, non-HIV, treatment naïve and non-cirrhotic); likewise, there is extra cost in checking for resistance mutations at baseline for patients with GT1a HCV assigned to receive EG.

6. PLOS authors have the option to publish the peer review history of their article (what does this mean?). If published, this will include your full peer review and any attached files.

Reviewer #1: No

---

## [Author Response · Author response to Decision Letter 0]

15 Feb 2021

We are grateful for your time and kind efforts to review our manuscript and for providing us with your invaluable comments to improve our manuscript. We have carefully addressed each issue and revised the manuscript accordingly as individually described below.

and

 Response: We have checked that our manuscript meets the requirement and format written in the above document and confirmed them after some corrections.

'I have read the journal's policy and the authors of this manuscript have the following competing interests:

Isao Sakaida received a Lectures fee from Gilead. Isao Hidaka received a Lectures fee from Gilead and Abbvie.'

a. Please confirm that this does not alter your adherence to all PLOS ONE policies on sharing data and materials, by including the following statement: "This does not alter our adherence to PLOS ONE policies on sharing data and materials.” (as detailed online in our guide for authors http://journals.plos.org/plosone/s/competing-interests). If there are restrictions on sharing of data and/or materials, please state these.

Please note that we cannot proceed with consideration of your article until this information has been declared.

Please know it is PLOS ONE policy for corresponding authors to declare, on behalf of all authors, all potential competing interests for the purposes of transparency. PLOS defines a competing interest as anything that interferes with, or could reasonably be perceived as interfering with, the full and objective presentation, peer review, editorial decision-making, or publication of research or non-research articles submitted to one of the journals. Competing interests can be financial or non-financial, professional, or personal. Competing

interests can arise in relationship to an organization or another person. Please follow this link to our website for more details on competing interests:

http://journals.plos.org/plosone/s/competing-interests

Response: We updated the Competing Interests statement in the cover letter correctly as follows. “Isao Hidaka received honoraria from AbbVie and Gilead, and received research funding from AbbVie. Isao Sakaida received honoraria from Gilead, and received research funding from AbbVie. These don’t alter our adherence to PLOS ONE policies on sharing data and materials.” 

 Response: We followed the instructions and added the caption "Supporting information" in the end of the manuscript. 

L. 709〜 713

“Supporting information

S1 Table. Meta-regression algorithm.

S1 Fig. Estimated incidence of liver cirrhosis.

S1 Appendix. Structure of the Markov model and stage-specific fibrosis progression rates in the natural history model of chronic hepatitis C virus infection.”

Reviewer #1: Since the first direct-acting antivirals (DAAs) received US Food and Drug Administration approval in 2011, several cost-effectiveness investigations have compared DAA-based regimens to previous standard-of-care regimens to calculate ICERs. They have also investigated the cost-effectiveness of eliminating HCV treatment restrictions. Compared to interferon-based regimens, the ICER for DAAs has consistently been estimated at <$100,000 per QALY for all genotypes and fibrosis stages.

Several studies have compared DAA regimens against one another. In general, when given a choice between recommended HCV DAA regimens, the less costly regimen is preferred as a more efficient use of resources

(even if it requires multiple tablet dosing). Because of the similar efficacy of most DAA regimens, cost becomes the critical factor driving relative cost-effectiveness. Studies have also estimated the cost-effectiveness of HCV treatment in special populations, including patients awaiting liver transplantation, HIV/HCV-coinfected patients,

those with chronic kidney disease, persons who inject drugs, and adolescents—all with favorable ICERs. At this time, it is reasonable to conclude that DAA regimens provide good value for the resources invested. That is the reason why AASLD and EASL guidelines have endorsed a treat all approach.

That being said, the analysis presented here has relevance due to 1) its focus on the Japanese population with GT1 HCV and 2) tweak association between willingness-to-pay and the real-world bottom line.

Societal willingness-to-pay thresholds in CEAs are not based on actual budget calculations and have little relationship to a payer’s bottom line. Under these premises, the results of the study presented here are important to give Japanese policy makers estimates of the cost of eliminating HCV. This is a timely study as Japan just this week has released its HCV elimination plan. Please consider adding these elements to the background section. The study objective goes beyond the simple verification of ICER below $50,000 to $100,000, by providing contemporary cost perspective to national elimination plans underway.

 Response: According to the issues the reviewer pointed out, we added following information to the background as follows:

1. Regarding the national HCV elimination plan.

L. 38〜39 

“The 69th World Health Assembly in 2016 adopted the first ‘Global Health Sector Strategy on Viral Hepatitis’ to eliminate viral hepatitis by 2030 as a public health threat. [2]” 

L. 45〜47 

“Therefore, comprehensive measures to combat hepatitis have been implemented including the public subsidy program for hepatitis treatment, which covers newly approved antiviral agents even in Japan.”

L. 70〜71 

“As a result of these measures, viral elimination has been successfully progressing [11, 12].”

2. Regarding previous studies showing evidence of a favorable cost-effectiveness of DAA treatment for patients with HCV irrespective of genotype and comorbidities, and the cost burden of DAA treatment under the universal health insurance coverage:

L. 72〜88

“On the other hand, facing the cost burden under the universal health coverage system, we have to reveal the cost-effectiveness of the DAA treatment for HCV-infected patients without contraindications [13].

Several studies that confirmed the cost-effectiveness of the DAA treatments for HCV-infected patients were published in other countries [14]. Besides, previous studies have also revealed the favorable cost-effectiveness of DAA treatment for the patients regardless of their genotype [15], adolescent patients [16], patients with HIV infection [17], and the universal screening program for HCV followed by DAAs treatments for the general population and subpopulations including prisoners and injecting drug users [18]. These results led to the recommendations by the AASLD and EASL and should guide the judgments of the health policymakers. Remarkably, the latest DAA regimens have a similar profile of effectiveness and safety, therefore, a difference in their costs mostly impacts the cost-effectiveness, and these results would give some evidence for the selection of the specific DAA treatment.

So far, however, we could not find any studies evaluated the cost-effectiveness of the DAA treatment for all patients with HCV infection, including those of less or no fibrosis under Japanese circumstances.”

3. Regarding our aim of this study, and also providing policymakers the information of cost-effectiveness of DAA treatment which one measure of the HCV elimination

L. 96〜 99

“This study's object was to reveal the cost-effectiveness of these DAAs treatment for all treatment naïve patients with chronic hepatitis C genotype 1 irrespective of their fibrosis stage, to provide the policymakers with the cost perspective of national measures toward HCV hepatitis elimination.”

Minor revisions:

1. The segment below need a few linguistic corrections. Fig 2 Treatment strategies 180 Infection to Death arrow is shown HCV natural history. and dot is shown reflecting progression 181 of fibrosis stages as well as decompensated liver cirrhosis and hepatocellular carcinoma.

182 Comparisons were made with 6 strategies that is 5 timing treatment strategy and no treatment 183 strategy.

 Response: Due to a typing error, we revised the manuscript to delete an unnecessary dot after "history." Also, the word "6 strategy" was singular, so we fixed it to be grammatically correct as follows. 

L. 184〜 187

 “From Infection to Death, an arrow is shown for HCV natural history, and a dot is shown reflecting the progression of fibrosis stages as well as decompensated liver cirrhosis and hepatocellular carcinoma. Comparisons were made with 6 strategies comprising of 5 timed treatment strategies and a no treatment strategy.”

2. Table 2 in line 191: needs formatting as follows: a) table title needs to be more specific, as “treatment effects” says little about the information covered by the table; b) abbreviations such as CH and LC need to be translated in the bottom of the table (chronic hepatitis? Liver cirrhosis?); c) the footnote statement “†Including

the previously treated patients. (N=171)” seems confusing to me, as I do not see 171 patients in the SL trial, only 83 patients. If there is another SL study in Japanese populations, I suggest including this study too so the table can show all 171 patients as stated.

Response: The title of Table 2 has been changed as follows to reflect the content precisely.

L. 196

“Baseline characteristics and main outcomes of the phase 3 studies of DAAs.”

We have also added annotations below the table for abbreviations. 

L. 198

“CH: chronic hepatitis; LC: compensated cirrhosis; SVR: sustained virologic response.” 

Regarding the number of enrolled patients in the phase 3 trial of SL, 83 were treatment-naïve patients and 88 were previously treated, and the total number of patients was 171. The SVR rate and discontinuation due to side effect data corresponded to 83 treatment-naïve patients, but we could only obtain the figures of age and gender data for all patients, including treatment-naïve and previously treated ones. Therefore, we added the following text to the annotations.

L. 199〜 200

“†N=171: Due to the lack of separated data on the age and male ratio of treatment-naïve patients, these values were the average of 171, patients including the previously treated patients.” 

3. Table 3 line 214: in regards to the cost of each treatment regimen, many patients receiving SL can be treated for 8 weeks only (viral load less than 6 million, non-black, non-HIV, treatment naïve and non-cirrhotic); likewise, there is extra cost in checking for resistance mutations at baseline for patients with GT1a HCV assigned to receive EG.

Response: 1) As the reviewer mentioned above, some cases might be treated for a shorter time. However, current Japanese guidelines for hepatitis C treatment recommend 12 weeks of treatment for patients with HCV genotype 1 for both SL and EG, 8 weeks for chronic hepatitis, and 12 weeks for cirrhosis for GP. Therefore, we analyzed our data using these treatment durations. We also examined whether the eight-week treatment with SL, as the reviewer mentioned above, lead to considerably different results compared to the original one. However, such a short treatment duration was more favorable for an earlier commencement of SL treatment, because the initial cost burden of SL decreased greatly.

We have added the following sentence to the text.

L. 191〜 193

“The duration of treatment was 12 weeks except for patients with non-cirrhosis CH receiving GP, in which it lasted 8 weeks based on the Japanese guidelines [30].”

2) 

We presumed that physicians do not measure resistance mutations before DAA treatment for treatment-naive patients in Japan, because national insurance does not cover it so far, and the current Japanese guidelines for DAA treatment in HCV infection do recommend the determination of mutations for treatment-naïve patients before treatment.

We additionally evaluated the cost impact of measuring resistance mutations on the cost-effectiveness as follows, and confirmed that ICER between TA and F1 did not worsen considerably, therefore not changing our conclusion.

Finally, we have added the following statement to the manuscript to clarify this as one of the assumptions. 

L 247〜L 248

“6. We did not consider analyzing mutations, as they were not covered by the national insurance system and performed routinely in current practice for treatment-naïve patients.”

---

## [Editor Report · Decision Letter 1]

5 Mar 2021

Cost-effectiveness of a “treat-all” strategy using direct-acting antivirals (DAAs) for Japanese patients with chronic hepatitis C genotype 1 at different fibrosis stages

PONE-D-20-30605R1

Dear Dr. Suenaga,

We’re pleased to inform you that your manuscript has been judged scientifically suitable for publication after your complete response to the review's comments and will be formally accepted for publication once it meets all outstanding technical requirements.

Kind regards,

Isabelle Chemin, PhD

Academic Editor

PLOS ONE
---

## [Editor Report · Acceptance letter]

12 Mar 2021

PONE-D-20-30605R1 

Cost-effectiveness of a “treat-all” strategy using direct-acting antivirals (DAAs) for Japanese patients with chronic hepatitis C genotype 1 at different fibrosis stages 

Dear Dr. Suenaga:

I'm pleased to inform you that your manuscript has been deemed suitable for publication in PLOS ONE. Congratulations! Your manuscript is now with our production department. 

Kind regards, 

on behalf of

Mrs Isabelle Chemin 

Academic Editor

PLOS ONE